# Technical Note: Including hydrologic impact definition in climate projection uncertainty partitioning: a case study of the Central American mid-summer drought

Edwin P. Maurer[1], Iris T. Stewart[2]

[1]Civil, Environmental, and Sustainable Engineering Department, Santa Clara University, Santa Clara, CA 95050, USA
[2]Environmental Studies and Sciences Department, Santa Clara University, Santa Clara, CA 95050, USA

*Correspondence to*: Edwin P. Maurer (emaurer@scu.edu)

**Abstract.** The Central American mid-summer drought (MSD) is a defining precipitation pattern within the regional hydrologic system linked to water and food security. Past changes and future projections in the MSD show a strong sensitivity to how the MSD is defined. The question then arises as to whether multiple definitions should be considered to capture the uncertainty in projected impacts as climate warming continues and a need to understand the impacts on regional hydrology persists. This study uses an ensemble of climate models downscaled over Nicaragua using two methods, global warming levels up to +3 °C, and different definitions of the MSD to characterize the contributions to total uncertainty of each component. Results indicate that the MSD definition contributes the least to total uncertainty, explaining 5-9% of the total. At the same time, evidence suggests a shift of the MSD to later in the year. As warming progresses, total uncertainty is increasingly dominated by variability among climate models. While not a dominant source of uncertainty, downscaling method adds approximately 8-18% to total uncertainty. Future studies of this phenomenon should include an ensemble of climate models, taking advantage of archives of downscaled data to adequately capture uncertainty in hydrologic impacts. These findings provide critical guidance for future research aiming to inform water planning and adaptation efforts in the region: by identifying the dominant sources of uncertainty across warming levels, this framework helps prioritize where to focus modeling and monitoring efforts. In particular, water resource managers can use this information to design adaptive strategies that are robust to model spread and shifts in seasonal precipitation timing, rather than to definitional ambiguity. The projection uncertainty partitioning approach could serve as a template to quantify the relative importance of uncertainty for projections of other precipitation-driven phenomena in different geographic contexts.

## 1 Introduction

Central America is consistently identified as a global hotspot for anthropogenic climate change, being prone to exacerbated impacts of already considerable natural climate variability and change (e.g., Giorgi, 2006; Hidalgo et al., 2017; Stewart et al., 2021). Any effort to develop strategies for mitigating impacts of future climate disruption or to adapt to probable hydrologic

impacts is based on climate model projections (IPCC, 2023; Lemos and Rood, 2010; Zhao et al., 2021). A quantitative assessment of how variability in precipitation is partitioned into other hydrologic processes, especially the evaluation of changes in extremes such as droughts and floods, can help anticipate variability in impacts (Yin and Roderick, 2020).

This study focuses on future precipitation-driven hydrologic changes, which introduce a cascade of uncertainties into impact

projections (Aitken et al., 2023). The uncertainty associated with each step along this cascade, which can include future greenhouse gas concentrations, climate response, downscaling, and hydrologic response can be estimated using multi-model ensembles (discussed in more detail below). Assessing this uncertainty can be a daunting task for stakeholders preparing strategies to cope with the projected changes in the timing and availability of water. Improved understanding of the comparative magnitudes of different sources of variability in impact projections can highlight opportunities to reduce them.

Even more importantly, these comparative magnitudes can help identify which steps in the modeling chain may be simplified without adversely affecting metrics relevant to decision-making related to adaptation and mitigation strategies in water resources (Steinschneider et al., 2023).

As characterized by early efforts to compare variability among precipitation and temperature predictions (Hawkins and

Sutton, 2009, 2011), uncertainties arise from imperfect representation of the earth system in numerical models (scientific uncertainty), the inability to know future atmospheric concentrations of greenhouse gases (forcing or scenario uncertainty), and the impossibility of precisely predicting the behaviour of a chaotic system (internal variability). Hawkins and Sutton found that, using climate model projections from the third Coupled Model Intercomparison Project (CMIP3), internal variability becomes less important than scientific or scenario uncertainty later in the 21st century. They also observed a

marked difference between precipitation projections, with greater internal and model variability persisting late into the 21st century, and temperature projections, which showed scenario uncertainty dominating projections in most regions late in the 21st century. This reflects the dominant physics of temperature being a primary response to the increased radiative forcing of accumulating greenhouse gases, and precipitation being driven by secondary physical processes that are more challenging to model, such as the moisture holding capacity of the atmosphere, the variety of phenomena that can cause precipitation, and

feedbacks with the land surface, ocean, and cryosphere lead to significant variability on scales much smaller than those of temperature (Neelin et al., 2022; O'Gorman and Schneider, 2009; Stainforth et al., 2005). Other studies have found similar results with more recent climate model simulations at continental scales (Lehner et al., 2020; Woldemeskel et al., 2016).

In the most recent sixth assessment report of the Intergovernmental Panel on Climate Change (IPCC), a new emphasis was

placed on assessing impacts at specified levels of global warming (relative to pre-industrial conditions of 1850–1900) to facilitate comparisons with earlier reports and coordination with targets in international agreements (IPCC, 2023). Assessing impacts at specific global warming levels also allows the use of models irrespective of their sensitivity (Hausfather et al., 2022). This approach essentially combines scientific and scenario uncertainties into a single 'projection' uncertainty,

reducing the variability in simulated projections, but leaving the time at which any specified level of warming occurs less well defined. An advantage for stakeholders is that policies can be developed to respond to locally important hydrologic impacts at different levels of warming without having to cope with forming an ensemble by culling models (based on correspondence of model sensitivity to a likely range) or with selecting atmospheric greenhouse gas concentration scenarios (Merrifield et al., 2023). In fact, demonstrable skill may be lost when excluding models from an ensemble based solely on correspondence of model sensitivity to observational estimates (Goldenson et al., 2023; Swaminathan et al., 2024).

Because hydrologic impacts analysis often requires projections at a finer spatial scale than what climate models produce, some type of downscaling is performed, which adds an additional layer of uncertainty that has been included in more recent studies (Lafferty and Sriver, 2023; Michalek et al., 2024; Wootten et al., 2017). The selection of downscaling method has been found in some locations to add a significant amount of uncertainty to projections, sometimes persisting at levels comparable to other sources through the 21st century, though results can vary widely in different regions (Lafferty and Sriver, 2023; Wootten et al., 2017).

When expanding an analysis to include specific impacts, varying definitions of impacts will add to the total uncertainty. For example, for future projections of potential evaporation (PE) for France, Lemaitre-Basset et al. (2022) found the PE formulation had a minor contribution to total projection uncertainty, except when only a single scenario was used. How droughts were characterized for compound hot and dry events was a dominant uncertainty source for low precipitation events but was a much smaller portion of uncertainty for other formulations (Jha et al., 2023). Even when given identical input, different models will simulate different impacts, compounding the uncertainty in projections (Chegwidden et al., 2019; Clark et al., 2016). The importance of this level of uncertainty can vary widely, based on the specific impact assessed (Bosshard et al., 2013).

Across Central America, the midsummer drought (MSD) is a phenomenon where boreal summer seasonal rainfall is characteristically divided into two distinct rainy periods by a relative lull in precipitation, and it is a critical component of the regional hydrologic system (Anderson et al., 2019). Changes in the MSD can lead to lower soil moisture, reduced groundwater recharge, and increased evaporation rates, which can have important impacts on the agricultural calendar, and local food and water security (Stewart et al., 2021). Thus, understanding the causes and impacts of the disruption of MSDs is crucial for managing water resources, predicting agricultural outcomes, and mitigating the effects of such dry periods. A recent study of the Central American MSD explored the variability in historical trends based on how the MSD is defined (Maurer et al., 2022). In addition, many studies have examined projected future changes in the MSD (Corrales-Suastegui et al., 2020; Maurer et al., 2017; Rauscher et al., 2008), though whether the uncertainty added by the MSD definition is important relative to other projection uncertainties remains to be determined and is the focus of this study.

The Central American Dry Corridor (CADC) is a highly climate sensitive region that occupies much of the Pacific side of Central America. The CADC is generally dry and has highly seasonal and variable climatic conditions, one expression of which is the MSD. The MSD persists across much of the region, strongly influencing smallholder farmers who depend on rainfed agriculture (Stewart et al., 2021). In Nicaragua, distinctly precarious socio-economic and climatic vulnerabilities intersect with a scarcity of observational (station) data (Girardin, 2024), rendering advances in the understanding of the regional hydrologic system particularly pertinent (Stewart et al., 2021).

In this study, we demonstrate a method of uncertainty partitioning for the MSD in Nicaragua to determine whether the choice of MSD definition is important to include as an additional source of uncertainty when estimating projected future impacts. We also recast the typical uncertainty analysis using specific warming levels rather than defined time windows so the results will be less sensitive to changes in the models selected or future emissions scenarios in projecting impacts, in this case to MSD characteristics.

## 2 Methods

The main sources of uncertainty in projections of the mid-summer drought (MSD) evaluated in this study include internal variability, differences among climate models, the choice of downscaling method, and the definition of the MSD itself. They are determined based on climate projections of daily precipitation and the simulated MSD characteristics (the impacts of concern for this study) and described as follows.

### 2.1 Climate model projections

Downscaled daily precipitation data are obtained from two sources: the climate impacts lab (CIL) data set (Gergel et al., 2023), and the CMIP6 version of the NASA-NEX archive (Thrasher et al., 2022). While both data sets use statistical downscaling, their methods are distinct. CIL uses a quantile delta mapping method for bias adjustment (Cannon et al., 2015) with a downscaling method that preserves climate model trends at quantiles. The NASA-NEX data set uses a similar bias correction, but a very different spatial disaggregation method based on perturbing the historical observations with bias corrected anomalies, without preserving precipitation trends of the climate models. Additionally, the two methods use different observational baselines for bias correction, which has been shown to influence results (Rastogi et al., 2022; Wootten et al., 2021). Both the NASA-NEX and CIL downscaled data have a resolution of 0.25 degrees (approximately 27.5 km in Nicaragua). It should be noted that restricting the analysis to model runs common to both CIL and NASA-NEX, with a single run per model, may limit the characterization of internal variability by relying on single realizations per model, and equal model weighting may understate the effect of model dependence or skill.

This study uses a set of eight climate model runs that are shared between both data sets for both historic and future projections, using shared socioeconomic pathway (SSP) 5-8.5 (Meinshausen et al., 2020). These are listed in Table 1, which

also includes the original spatial resolution before downscaling. SSP5-8.5 is the scenario with the highest anthropogenic emissions and resulting radiative forcing, which means all the models used in this study produce in excess of +3 °C of warming during the 21st century, allowing all to be used in analyses at global warming levels of +1.5, +2.0, and +3.0 °C.

**Table 1: Climate model runs used by downscaling methods in this study. Nominal resolution is the approximate horizontal**
**resolution of the archived data for the model land component.**

| Model | Variant | Institution | Nominal Resolution (km) |
|---|---|---|---|
| BCC-CSM2-MR | r1i1p1f1 | Beijing Climate Center | 100 |
| CMCC-ESM2 | r1i1p1f1 | Euro-Mediterranean Center | 100 |
| EC-Earth3-Veg-LR | r1i1p1f1 | EC-EARTH consortium, The Netherlands/Ireland | 250 |
| GFDL-ESM4 | r1i1p1f1 | NOAA Geophysical Fluid Dynamics Laboratory | 100 |
| INM-CM5-0 | r1i1p1f1 | Institute for Numerical Mathematics (INM), Russia | 100 |
| MIROC6 | r1i1p1f1 | National Institute for Environmental Studies, Japan | 250 |
| MPI-ESM1-2-HR | r1i1p1f1 | Max Planck Institute for Meteorology (MPI), Germany | 100 |
| NorESM2-MM | r1i1p1f1 | Norwegian Climate Center, Norway | 100 |

By only including those downscaled runs that use identical climate model simulations for both CIL and NASA-NEX as input, the variability due to model selection is separated from that due to internal variability represented by different model initial conditions or parameterizations. All model projections are considered equally plausible and are thus equally weighted

as in Michalek et al. (2023).

**2.2 Warming levels**

The years at which each model projection reaches +1.0, +1.5, +2.0, and +3.0 °C of global mean warming (relative to pre-industrial conditions) were determined by Hauser et al. (2022) for CMIP6 climate models using the mid-year of a 20-year moving window. In this experiment, a 30-year window was used around the defined mid-year for each model run and the

mean of each impact in that 30-year period was determined at different levels of warming. The years at which the model projections simulate the different levels of warming are shown in Figure 1. The +1.0 °C warming level is not used in this study as it has already been exceeded.

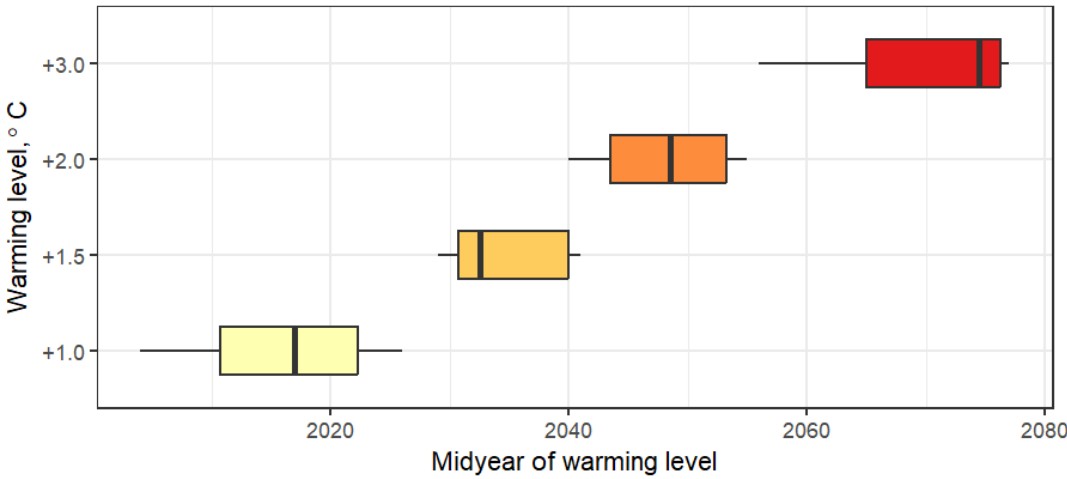

**Figure 1: The years at which each warming level is reached for the climate model ensemble in Table 1.**

 **2.3 MSD characteristics**

The MSD as a hydrologic phenomenon is defined using local stakeholder descriptions and the methods described in Maurer et al. (2022). The methods are implemented in the R package msdrought (Uyeda et al., 2024). The MSD characteristics are entirely derived from the timing and magnitude of smoothed daily precipitation, and the occurrence of two maxima and a relative minimum within two defined windows, as depicted in Figure 2. Years that do not display a timing of peaks and the
155 intervening minimum occurring within the defined windows are designated as NULL. The default definition for the Central American region requires that the MSD maxima must occur between May 1 and October 31 and the minimum within the June 1 to August 31 window (Figure 2), though these are adjusted for this experiment as noted below.

Whether an MSD occurs in any year is often defined using some measures of duration, intensity (or strength), and timing
(e.g., Alfaro, 2014; Anderson et al., 2019, Karnauskas et al., 2013; Perdigon-Morales, et al., 2018). Maurer et al. (2022) determined that, considering several aspects of MSD definition, the two with greatest impact on results were the minimum intensity and the MSD timing, which are therefore used in this study. For this experiment, we use the same definitions of intensity and duration as Maurer et al. (2022): MSD intensity was calculated as the mean precipitation of these maxima minus the minimum precipitation occurring between them; MSD duration was defined as the number of days between the
two seasonal precipitation maxima. To explore the variability associated with MSD definitions, the dates are shifted 14 days earlier and then 14 days later from those in Figure 2 to estimate the effect of this definition on MSD variability. We also vary the minimum intensity from 2 to 4 mm d$^{-1}$.

While not a defining characteristic, the frequency of MSD occurrence is often used to characterize the robustness and importance of the MSD (Corrales-Suastegui, et al., 2020; Zhao et al., 2023). Where we present results, we focus on regions that exhibit MSDs in ≥ 50% of years.

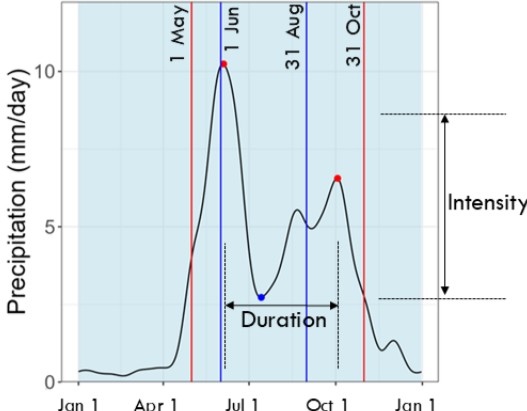

**Figure 2: A schematic of a typical MSD year, highlighting the definition (dates) and the impacts (intensity and duration) of interest in this study. Red points mark maxima and the blue point marks the minimum used. Duration is the number of days between the peaks; intensity is the average of the two peaks minus the minimum between them. All metrics are calculated from the smoothed daily precipitation time series. To estimate the effect of the definition on MSD variability, dates are shifted 14 days earlier and later from the definition dates shown here, and differing values for a minimum intensity are applied.**

### 2.4 Variance partitioning

The partitioning of variance among the different sources generally follows Michalek et al. (2023). Variance partitioning was done for each MSD characteristic/impact (duration and intensity), for each grid cell in the domain bounded by longitudes -83° and -88° and latitudes 10° and 15°.

First, for each climate model, downscaling method, and definition (the experiments varying the MSD dates and minimum intensity) an 11-year smoothing window was applied to the values for each year and the anomalies relative to a 1970-1999 base period were calculated. Internal variability was then estimated by fitting a LOESS curve to the anomalies of each impact and calculating the variance of the LOESS residuals for each defined warming level and impact of interest (+1.5, +2.0 and +3.0 °C) using a 30-year window centered on the midyear of warming for each climate model. Some prior studies have used other methods to estimate internal variability, such as fitting a polynomial rather than a LOESS curve (Hawkins and Sutton, 2009). The choice of method for estimating internal variability has been shown to add substantial uncertainty when a single climate model is used (with many runs); using multiple climate models lessens this impact (Lehner et al., 2020).

Model variability is estimated by calculating the variance of the LOESS predicted values for each defined warming level and impact of interest.

$$\text{Model Variance} = \frac{1}{N_1}\sum_{d,e} \text{var}[\hat{x}(t,d,e,m)] \tag{1}$$

Here $\hat{x}$ is the set of LOESS predicted values for the set of years, $t$, associated with the warming level specified for each climate model, $m$. $N_1$ is the number of unique subsets of $\hat{x}$ with valid (non-NULL) MSD impact data for each combination of downscaling method, $d$, and MSD definition experiment, $e$. Similarly, uncertainty due to the downscaling method is calculated by

$$\text{Downscaling Variance} = \frac{1}{N_2}\sum_{m,e} \text{var}[\hat{x}(t,d,e,m)] \tag{2}$$

where $N_2$ is the number of unique subsets of $\hat{x}$ with valid (non-NULL) MSD impact data for each combination of climate model, $m$, and MSD definition experiment, $e$. Finally, the uncertainty due to MSD definition is calculated by

$$\text{MSD Definition Variance} = \frac{1}{N_3}\sum_{m,d} \text{var}[\hat{x}(t,d,e,m)] \tag{3}$$

$N_3$ is the number of unique subsets of $\hat{x}$ with valid (non-NULL) MSD impact data for each combination of downscaling method, $d$, and climate model, $m$.

## 3 Results and Discussion

To frame the impacts of a warming climate on the MSD in Nicaragua, Figure 3 shows the median changes in intensity and duration projected by the complete ensemble used for this study. Figure 3 also shows the boundary of the Central American Dry Corridor (CADC) as objectively determined by Stewart et al. (2021). The CADC is a relatively arid region with highly seasonal precipitation in Central America that exhibits a high sensitivity to climatic changes and is especially susceptible to drought impacts. It is therefore a focus for some of the analysis in this study.

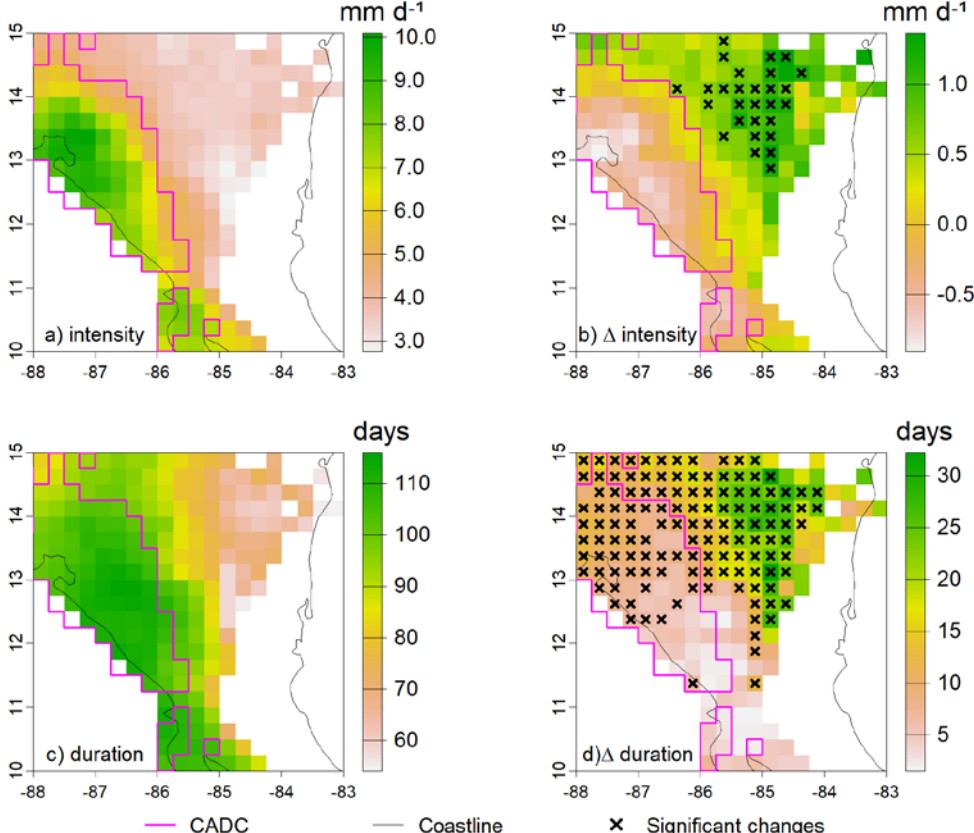

**Figure 3: Historical values as simulated by the model ensemble of MSD intensity (a) and duration (c) for 1970-1999, and the projected changes (b and d) with +3 °C of global warming, using the dates in Figure 2 in the MSD definition. Grid cells marked with an "X" indicate the change is significant at a 5% level based on a Wilcox (Mann-Whitney) test. The magenta line is the boundary of the Central America Dry Corridor (CADC), the black line denotes the coastline. Grid cells with less than 50% of years having an MSD, in both historical and future periods, are white. In addition, if less than half of the models in the ensemble show an MSD, the grid cell is white.**

Figure 3 shows the highest intensity and longest duration MSDs for 1970-1999 are experienced in the CADC on the Pacific side of Nicaragua. Changes in MSD intensity anticipated with +3 °C of global warming are focused on the East, in the area that has historically experienced the lowest intensity (least pronounced) events of the shortest duration. Duration changes are more widespread, indicating a longer lull in the rainy season as climate disruption progresses.

Figures 4 and 5 show the effect of shifting the default dates in the MSD definition (Figure 2) on projected changes to MSD intensity and duration. Figure 4 shows that shifting the dates 14 days earlier dramatically reduces the area that would be classified as having an MSD, compared to Figure 3. Conversely, Figure 5 shows that shifting the time windows 14 days later expands the area with an MSD. These results are consistent with prior work that found the MSD tending to shift later and to have a longer duration with climate change impacts (Maurer et al., 2022).

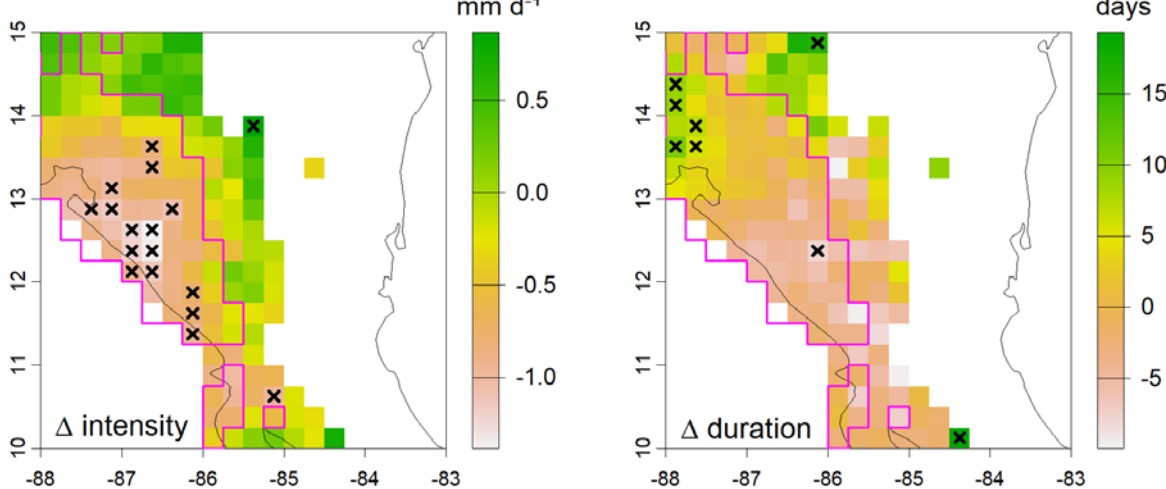

**Figure 4: As in Figure 3 but showing only mean changes in MSD intensity and duration when changing the MSD definition to use dates 14 days earlier than shown in Figure 2.**

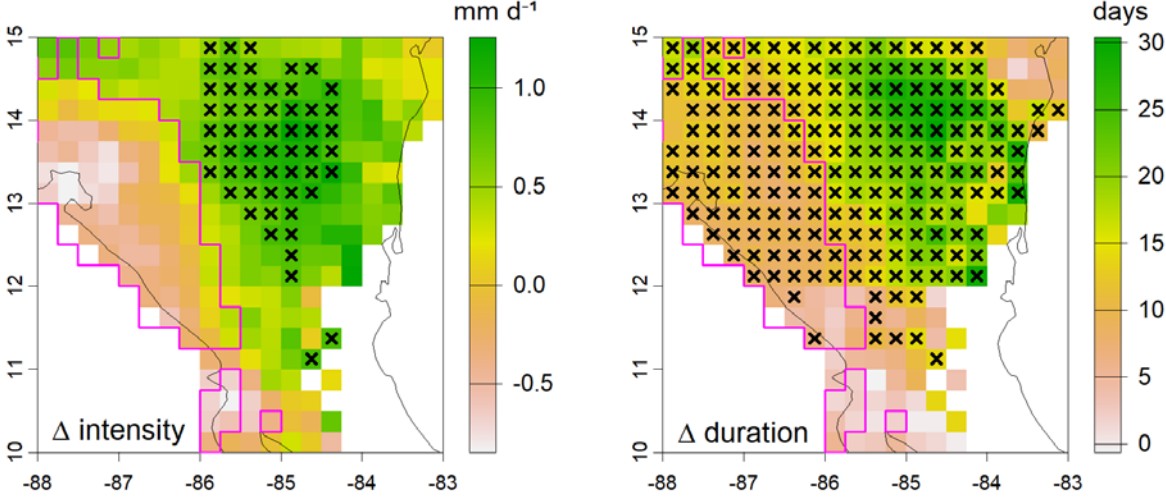

**Figure 5: Similar to Figure 4, but with the MSD definition using dates shifted 14 days later than in Figure 2.**

Focusing on the CADC in the Nicaragua domain considered in this study, Figure 6 shows the variability of the changes in MSD duration and intensity, averaged over the CADC, among the 16 different projections (eight climate models and two downscaling methods) when shifting the MSD definition dates 14 days earlier and 14 days later.

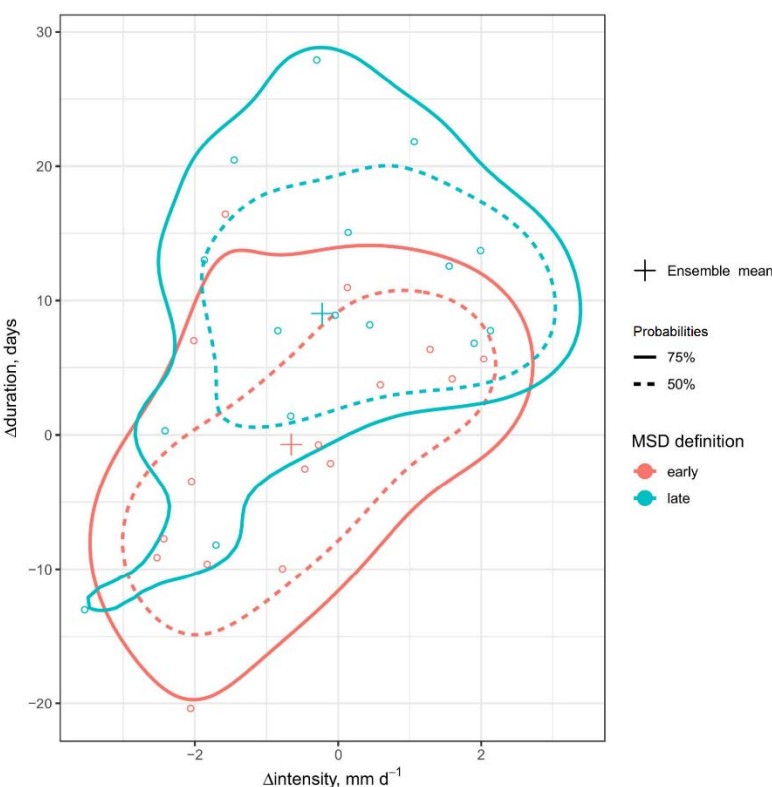

**Figure 6: Mean individual projections (points) for the CADC (Nicaragua) and probability contours (based on a Gaussian kernel density estimator) at +3 °C global warming with the MSD definition shifted 14 days earlier or 14 days later than in Figure 2.**

Figure 6 shows that for the CADC (the portion in Nicaragua) the definition of the dates has a strong impact on the projected changes, especially in MSD duration, with the shift in projected duration change being comparable to the variability among

245 individual projections. This raises the question of whether the choice of MSD definition adds enough uncertainty to the MSD impacts, relative to the other sources of uncertainty, where stakeholders should include multiple definitions in impacts analysis. This is explored below.

Figure 7 shows the contributions to total variance of MSD intensity due to the different sources considered in this study at

250 different global warming levels. As has been found in other analyses of precipitation uncertainties (Lehner et al., 2020; Wu et al., 2022), this precipitation-derived MSD impact also shows internal variability dominating for the historical period. Internal variability contributes a substantial amount of uncertainty as warming progresses, though even at the lowest +1.5 °C level the model projection uncertainty constitutes most of the uncertainty to the projection of MSD impact. While not shown, the MSD duration shows similar patterns.

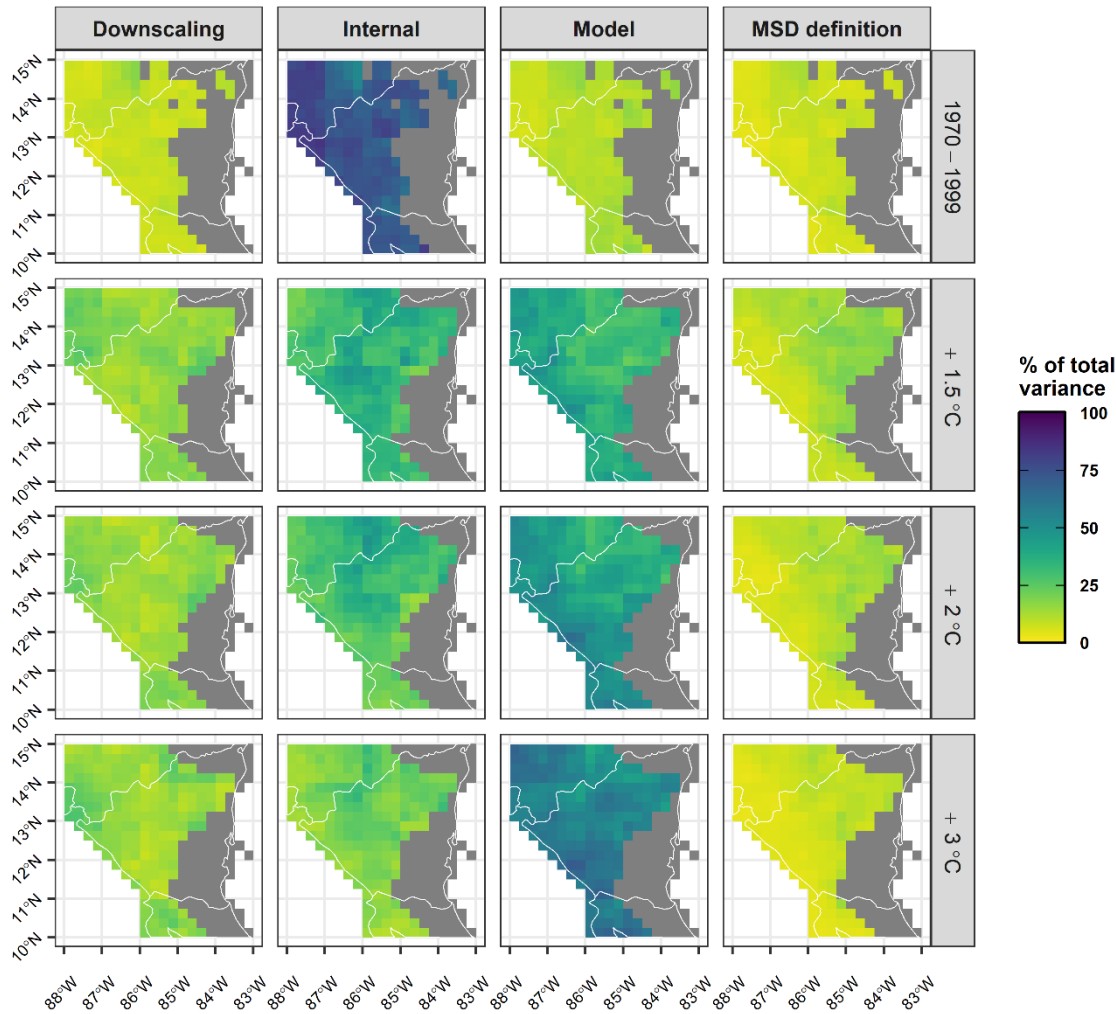

**Figure 7: For MSD intensity, the percent of total variance contributing to each source included in this analysis, for the base period of 1970-1999 (top row) and for different levels of global warming. Only grid cells exhibiting 50% or greater frequency of MSD years are colored.**

Despite very different spatial characteristics of changes in MSD intensity (Figures 3-5) and MSD frequency (Figure 8), Figure 7 shows relatively consistent fractional uncertainty for all sources across the domain. This reflects the larger contributions to MSD intensity uncertainty of climate model and internal variability, both inherited from the larger spatial scales of the climate models (Table 1). The bias correction and spatial downscaling included with the downscaling methods aligns the climate model output to finer gridded observations but adds a relatively small portion to the overall uncertainty in MSD impacts. As shown in Figure 7, uncertainty due to downscaling is relatively small over Central America because the dominant sources of uncertainty come from large-scale climate model differences and internal variability. Downscaling methods primarily refine model output to match finer observational data but do not significantly increase the total

uncertainty. Thus, in the context of MSD analysis, the role of downscaling in uncertainty is modest compared to inherited
uncertainties from the climate models themselves. The MSD definition has the smallest contribution to total uncertainty at all
warming levels with smaller contributions toward the Pacific coast where MSD frequency is greatest (Figure 8) and intensity
is strongest (Figure 3).

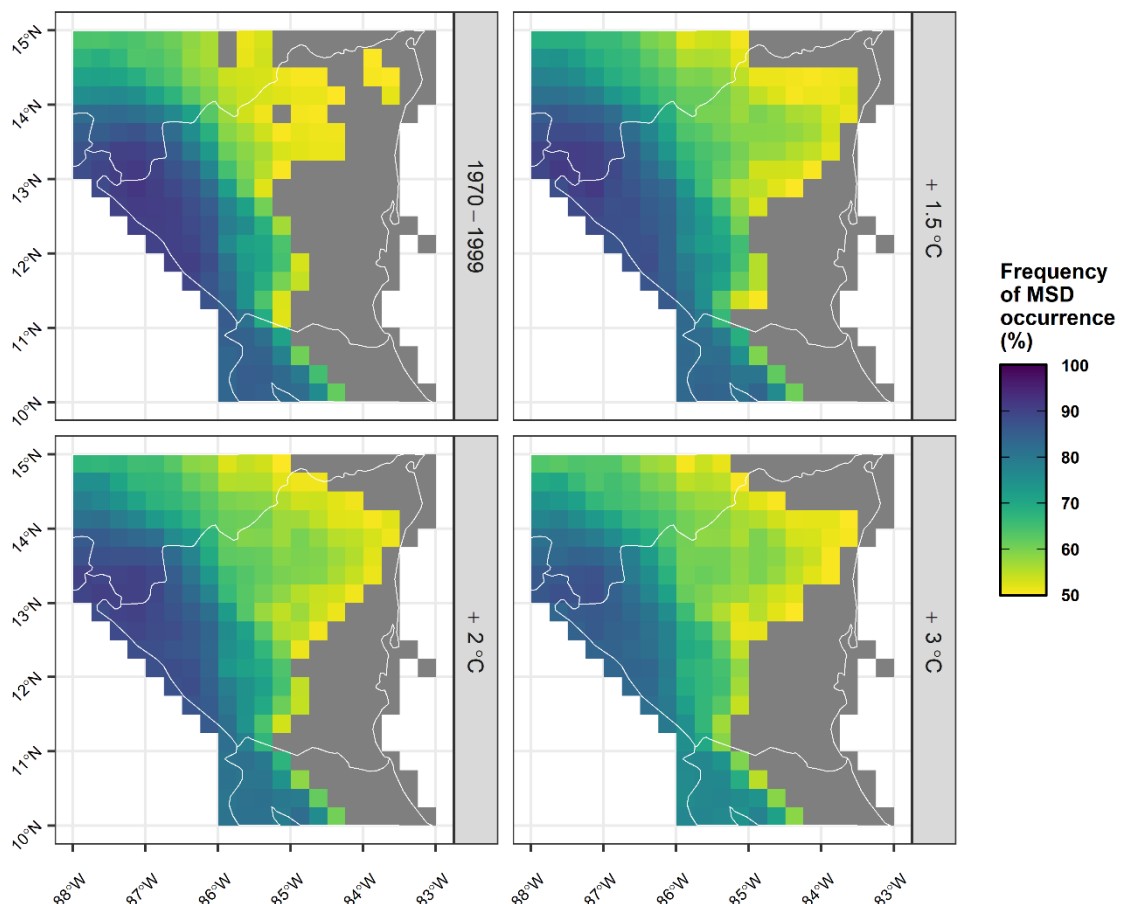

**Figure 8: The frequency of MSD occurrence as a % of years for the different global warming levels depicted in Figure 7, for**
**regions exhibiting an MSD in at least 50% of years.**

While MSD impacts in this study are based on (smoothed) daily precipitation, there might be more spatial heterogeneity in
impacts derived from extreme precipitation events, since mean daily precipitation is generally more skilfully simulated by
climate models than extremes (Volosciuk et al., 2017), and extremes would therefore be adjusted more dramatically during
the downscaling process. Exploring the uncertainty contribution of downscaling to impacts driven by more extreme events
may benefit from more varied, regionally-focused downscaling efforts (e.g., Tamayo et al., 2022).

The progression of uncertainty through different levels of warming for both MSD intensity and duration, averaged across the CADC in Nicaragua, is shown in Figure 9. At levels of warming above +2.0 °C model projection uncertainty is the largest component to uncertainty in both MSD impacts. At +3.0 °C of warming, internal variability contributes 19 – 29% of the total uncertainty in MSD intensity and duration over the CADC. Downscaling variability for the CADC region contributes a relatively consistent 8-18% of the total uncertainty at all future warming levels, and a larger percentage for MSD intensity than duration. This is consistent with different downscaling methods, which are often developed to adjust for biases in mean values (Cannon et al., 2015), diverging more for extreme precipitation, and MSD intensity being a function of peak precipitation values in any year.

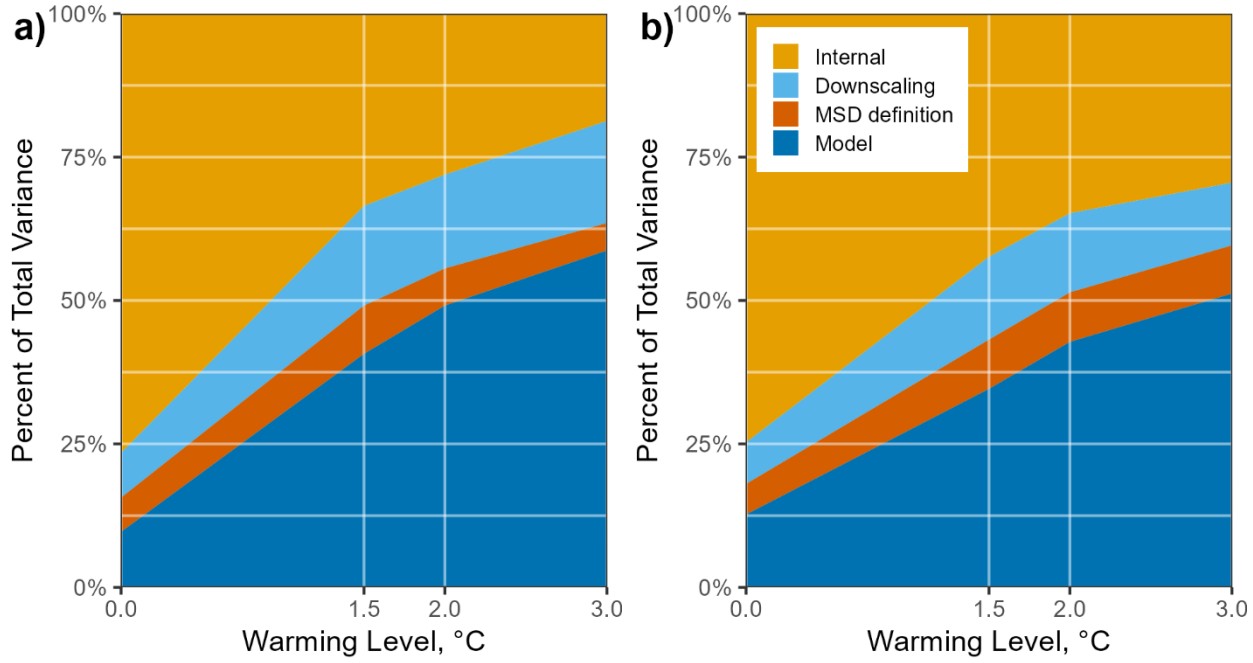

**Figure 9: Mean fraction of variance for a) MSD intensity and b) MSD duration, averaged over the CADC in Nicaragua (see Figures 4 and 5). The Warming level labeled 0 is for 1970-1999.**

The uncertainty due to the MSD definition is the smallest portion at all warming levels, at 5-9% of the total uncertainty over the CADC and is slightly larger for duration than for intensity as warming progresses. The uncertainty due to MSD duration becomes higher than that for intensity because for the CADC the intensities are already high, and even with projected slight declines in intensity (Figure 3), they remain well above the minimum thresholds explored in this experiment. By contrast, the timing established for the MSD windows has a dramatic effect on the determination of an MSD year and produces changes in duration that are large relative to its baseline (1970-1999) values (Figures 4 and 5). While these differences do emerge in our results, it should be emphasized that their contribution to total uncertainty remains small.

## 4 Conclusions

The advances in understanding the hydrologic system in this study focus on refining the methods for projecting future precipitation changes and their impacts on the Central American Midsummer Drought (MSD). In considering projections of future precipitation on the Central American MSD, this study indicates the dominant sources of uncertainty are internal variability (especially for near term, or lower levels of global warming) and variability among climate models (increasingly so as warming level increases). While precipitation downscaling has the potential to introduce large uncertainties in some hydrologic impacts, for the MSD impacts included in this study downscaling generally contributes less to total uncertainty compared to other sources. Despite having a strong impact on the magnitude and spatial extent of the MSD, the exact definition of the MSD has a minor effect on the uncertainty of MSD projections at all warming levels, similar to prior studies that found variable or limited impact of definition uncertainty (Jeantet et al., 2023; Jha et al., 2023; Lemaitre-Basset et al., 2022). Thus, while model spread and internal variability dominate, the role of the MSD definition was found in this study to be comparatively small; future studies should continue testing how event or season definitions influence uncertainty.

The main implication of these findings for future work on climate disruption and the future of the Central American MSD is that selecting an ensemble of climate models is essential for characterizing the uncertainty in precipitation and its impact on the MSD. By analysing impacts at specific levels of warming, rather than future spans of years, the selection of models may be done without excluding models based on sensitivity, which simplifies the process as other climate model skill metrics may be used. Using a single precipitation downscaling method for all climate models would still capture the majority of MSD impact uncertainty, though with multiple archives of downscaled data freely available, multiple methods can be readily included. The definition of the MSD can be chosen to capture impacts of interest, but the specific definition of the time windows and minimum intensity required for an MSD does not add substantially to the uncertainty in impacts. These findings show that future work can better support more efficient decision-making by selecting climate model ensembles based on performance metrics rather than sensitivity, framing projections by warming levels instead of time periods, and using a single downscaling method without major loss of uncertainty characterization. Additionally, they demonstrate that flexibility in defining the MSD allows tailoring analyses to stakeholder-relevant impacts without significantly affecting uncertainty. Furthermore, the dominance of internal variability at near-term timescales, and the growing role of model uncertainty under higher warming, has important implications for stakeholders. Policymakers, water managers, and agricultural planners must recognize that near-term variability may mask or amplify underlying trends, complicating adaptation strategies. In the longer term, the prominence of model uncertainty highlights the need for improved climate modeling and ensemble strategies to better constrain future risk assessments. Explicitly accounting for which source of uncertainty dominates at a given time horizon allows stakeholders to tailor their decisions accordingly.

While two precipitation downscaling methods were used to characterize the uncertainty in downscaling on the MSD impacts, including additional methods could improve this, especially if dynamic downscaling were represented. Expanding the domain would allow a greater exploration of the spatial variability in the different components of uncertainty on the regional MSD. Future research will further explore these improvements to this study. The approach presented in this study could serve as a template to quantify the relative importance on uncertainty for the projection of other precipitation-driven impacts in different geographic contexts and regional hydrologic systems, such as monsoon patterns or the timing and duration of the rainy season in other highly seasonal climates.

This study directly responds to the needs of stakeholders, such as water managers and agricultural planners, who require actionable and skilful projections to inform adaptation strategies under climate change. By identifying where simplifications in modeling (e.g., MSD definition or downscaling method) do not substantially impact uncertainty, and by adopting a warming-level framing that aligns with international policy targets, this work supports more efficient and targeted planning in the face of future hydrologic change that can be developed for other geographic regions.

## Code availability

An R package is available at https://cran.r-project.org/package=msdrought for determining the characteristics of the mid-summer drought using daily precipitation data. Processing code is archived at https://github.com/EdM44/msd_variance.

## Data availability

The NASA-NEX downscaled climate model output data are available at https://www.nccs.nasa.gov/services/data-collections/land-based-products/nex-gddp-cmip6. The CIL downscaled climate model output data may be obtained from https://planetarycomputer.microsoft.com/dataset/group/cil-gdpcir.

## Author contribution

EM and IS conceived of the project. EM conducted the analysis. EM and IS directed the development of the msdrought software. EM and IS prepared the original draft of the manuscript.

## Competing interests

The authors declare that they have no conflict of interest

**Acknowledgements**

We gratefully acknowledge support from the Whitham Foundation, the Miller Center at Santa Clara University, and the Environmental Justice and the Common Good Initiative at Santa Clara University. Our research partner in Nicaragua is CII-Asdenic in Estelí.

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
