# Peer review of "Technical Note: Including hydrologic impact definition in climate projection uncertainty partitioning: a case study of the Central American mid-summer drought"

_EGUsphere, 2025_

## Referee Comment (RC2)

**Review of HESS manuscript egusphere-2025-1650**

This manuscript analyzes uncertainty partitioning for the Central American midsummer drought (MSD) using multimodel ensembles, two downscaling datasets, and multiple MSD definitions evaluated at specific global warming levels. Its novel contribution is the explicit inclusion of MSD definition as an additional source of uncertainty, and it also demonstrates the value of applying a warming-level framing. This approach has been established in recent climate assessments but is applied here to hydrologic uncertainty partitioning to reduce sensitivity to model selection or emissions scenarios.

The study is a valuable contribution within the scope of HESS, addressing a climatically and socio-economically vulnerable region while applying an uncertainty framework relevant to decision-oriented science. The analysis is scientifically sound and the conclusions are well supported. With improvements to presentation clarity (streamlined writing, clearer figure formatting, consolidated summaries of uncertainty sources) and stronger connection to prior literature, the paper would make a strong contribution to HESS. After addressing the comments below, I would consider the scores across all criteria to be excellent.

**1. Scientific Significance**

The manuscript addresses a relevant and pressing hydrological issue in a vulnerable region and employs an uncertainty-partitioning framework aligned with HESS's interest in decision-relevant science. Its novel contribution is the explicit inclusion of the definition of the midsummer drought (MSD) as an additional quantified source of uncertainty, assessed alongside climate model spread, internal variability, and downscaling. The conclusion strengthens the broader relevance by noting that this framework can serve as a template for other precipitation-driven cases such as monsoons or rainy season timing (p.14, L285–287).

**Suggested enhancement**: The introduction notes that *"The uncertainty associated with each step…can become a daunting task for stakeholders"* (p.2, L30–33), but this theme is not revisited later. The discussion would be stronger if the authors explicitly linked their findings to future work aiming to be more decision-ready. In particular, they could emphasize two key contributions of their analysis: (i) that while model spread and internal variability dominate, the role of MSD definition is comparatively small, which highlights the need for future studies to continue testing how event or season definitions influence uncertainty; and (ii) that using specific warming levels rather than fixed time windows revealed how uncertainty evolves with global warming in a way that is less sensitive to model selection or emissions scenarios. Explaining why these findings are relevant for orienting future work toward decision-relevant applications would sharpen the paper's significance.

**2. Scientific Quality**

The study applies a sound methodological framework and delivers well-supported conclusions. However, certain aspects could benefit from deeper explanation or clarification:

- **Comparison with prior studies:**
  The introduction cites studies where impact definition uncertainty was substantial (e.g., Lemaitre-Basset et al., 2022; Jha et al., 2023; Jeantet et al., 2023). Since this study finds MSD definition is a minor contributor, the discussion or conclusions should briefly contrast this outcome with those earlier findings.
- **References:**
  Some statements require clearer sourcing. For example (L96–98), *"In Nicaragua, distinctly precarious socio-economic and climatic vulnerabilities intersect with a scarcity of observational (station) data, rendering advances in the understanding of the regional hydrologic system particularly pertinent"*. Stewart et al (2021) is cited in the prior sentence, but this is not a primary source for these specific claims. The authors should provide a more appropriate primary citation (e.g., to studies documenting data scarcity or socio-economic vulnerability in Nicaragua).
- **Methods:**
  Restricting the analysis to model runs common to both CIL and NASA-NEX is a sound way to separate downscaling differences from model selection effects. However, the characterization of internal variability may be limited by relying on single realizations per model, and equal model weighting may understate the effect of model dependence or skill. These limitations should be acknowledged.
- **Discussion depth:**
  - The results show downscaling adds ~10–15% uncertainty, but the discussion does not explain why this is relatively small compared to other studies. A short explanation (e.g., method characteristics, regional features) would help.
  - Figures 7–8 could be discussed in more depth, particularly why duration and intensity differ in uncertainty contributions and why MSD definition plays a larger role for duration.
  - Explicitly linking the dominance of model/internal variability back to implications for stakeholders would strengthen conclusions.
  - The introduction's explanation of why precipitation is harder to project than temperature is not revisited. Connecting this physical reasoning to the results would situate findings more broadly.

**3. Presentation Quality**

The manuscript is structured and purposeful overall, but several presentation enhancements would significantly improve readability and clarity:

- **Clarity of writing**:
  Some sentences are overly long or disrupted by parenthetical clauses, making them hard to follow.
    - For example, the sentence at lines 29–30 could be simplified to: "*This study focuses on future precipitation-driven hydrologic changes, which introduce a cascade of uncertainties into impact projections (Aitken et al., 2023).*"
    - Similarly, the sentence at lines 30–33 could be rephrased as: "*Uncertainty arises at each step of this cascade—including future greenhouse gas concentrations, climate response, downscaling, and hydrologic response—and can be estimated using multimodel ensembles (discussed in more detail below). However, this estimation can be daunting for stakeholders preparing strategies to cope with projected changes in the timing and availability of water.*"
    - Similar improvements can be made to other heavily clause-laden sentences.
- **Summary of uncertainties:**
  The uncertainty sources are described in text but not consolidated. A clear listing or table early in the manuscript would help readers track sources and enhance structural clarity.
- **Linking introduction to conclusions**:
  The conclusion would benefit from explicitly revisiting key themes from the introduction—stakeholders, simplification opportunities, warming-level framing—to improve narrative cohesion.
- **Definition of MSD metrics**:
  MSD intensity should be defined unambiguously, e.g., *"MSD intensity was defined as the mean precipitation of the two seasonal maxima minus the minimum precipitation between them (derived from the smoothed daily series)."* Figure 2 should clarify that red points mark maxima and the blue point marks the minimum used, ideally through a caption note and a simple legend.
- **Notation for warming levels:**
  Warming-level notation throughout the text and figures (Figures 1 and 8) should follow IPCC conventions, using a "+" sign before values (e.g., "+1.5 °C," "+2 °C," "+3 °C").
- **Abstract:**
  The abstract is informative, identifying which uncertainty sources dominate and provides guidance for future studies. To strengthen its impact, the authors could add a clearer statement of relevance for future work wishing to inform water planning/adaptation.
- **Figures**:
    - Ensure consistent font type and size (Arial or Helvetica recommended by HESS) across all figures.
    - **Figure 1:** Add "+" to Y-axis ticks; ensure "2080" is fully visible.

- **Figure 2:** Increase axis font size, add more x-axis ticks/labels, enlarge plotted blue and red points, clarify caption the caption to reference the blue and red data points, include a legend for the blue and red data points, fix cutoff of "0."
- **Figure 3:** Increase font size, and  include a legend for CADC boundary, coastlines, and significance marks.
- **Figure 6:** Spell out "Probability," remove underscore in "Ensemble_mean," improve font consistency.
- **Figure 7:** Make color bar clearer with black border/ticks to be consistent with color bars in other figures; replace the legend title of "Percent" with "% of total variance" or "Contribution to total variance (%)."
- **Figure 8:** Remove underscore from "MSD_definition," use a colorblind-friendly palette (e.g., Blue #0072B2, Orange #E69F00, Sky Blue #56B4E9, Vermilion #D55E00).

---

## Author Comment (AC1)

**Responses to reviewers - HESS manuscript egusphere-2025-1650**

We are grateful to the two reviewers for their careful reading and helpful comments on the submitted manuscript. We submit this response to these comments to show how we will revise the paper (or already have in some cases) by addressing each comment. Original comments are in regular type. Responses are in red and italics.

**Reviewer 1:**

This paper performs a variance decomposition analysis to identify important sources of uncertainty in projections of the Central American mid-summer drought (MSD). The novel contribution of this work is to include the definition of the MSD as an additional source of uncertainty. The paper is well-written, scientifically sound, and the motivation and framework have broad relevance across other impact areas. I have a few suggestions that I feel would improve the manuscript.

**Thank you for the positive feedback.**

**Main points:**

1. My main question concerns the apparent mismatch between Figures 4 & 5 and Figure 6. The distribution of white (meaning NULL) grid points in Figures 4 & 5 would suggest that the MSD definition has a considerable impact on its frequency and spatial extent, yet this doesn't show in the variance decomposition results of Figure 7. As I understand, the variance decomposition results for intensity and duration are conditional on the MSD occurring since the authors drop all NULL values, but I wonder whether a frequency-based metric might show qualitatively different patterns. As the authors are likely aware, frequency is commonly considered in tandem with intensity and duration, and often required to form a full understanding of impacts. Would the authors be able to repeat the analysis for a suitably-defined frequency metric? Or if not, sharpen the discussion of the apparent mismatch between Figures 4-5 and the decomposition results in Figure 7?

This observation highlights the difference between Figures 4 and 5 (and Figure 3), which each show a different part of the domain that has MSDs in  $\geq$ 50% of years, and Figure 7, which shows the variance partitioning for the entire domain, regardless of MSD frequency. We have revised the methods section to include the following:

"Whether an MSD occurs in any year is often defined using some measures of duration, intensity (or strength), and timing (e.g., Alfaro, 2014; Anderson et al., 2019, Karnauskas et al., 2013; Perdigon-Morales, et al., 2018). Maurer et al. (2022) determined that, considering several aspects of MSD definition, the two with greatest impact on results were the minimum intensity and the MSD timing, which are used in this study. For this experiment, we use the same definitions of intensity and duration as Maurer et al. (2022): MSD intensity was calculated as the mean precipitation of these maxima minus the minimum precipitation occurring between them; MSD duration was defined as the number of days between the two seasonal precipitation maxima. To explore the variability associated with MSD definitions, the dates are shifted 14 days earlier and then 14 days later from those in Figure 2 to estimate the effect of this definition on MSD variability. We also vary the minimum intensity from 2 to 4 mm d-1.

While not a defining characteristic, the frequency of MSD occurrence is often used to characterize the robustness and importance of the MSD (Corrales-Suastegui, et al., 2020; Zhao et al., 2023). Where we present results, we focus on regions that exhibit MSDs in  $\geq 50\%$  of years."

Revised caption: Figure 7: For MSD intensity, the percent of total variance contributing to each source included in this analysis, for the base period of 1970-1999 (top row) and for different levels of global warming. Only grid cells exhibiting 50% or greater frequency of MSD years are colored.

In addition, we include a new Figure 8 showing the frequency of the MSD throughout the domain, as shown below.

Figure 8: The frequency of MSD occurrence as a % of years for the different global warming levels depicted in Figure 7.

This figure is discussed in a revised paragraph:

"Despite very different spatial characteristics of changes in MSD frequency (indicated by the white grid cells in Figures 3-5) and MSD intensity (Figure 8), Figure 7 shows relatively consistent fractional uncertainty for all sources across the domain. This reflects the larger contributions to MSD intensity uncertainty of climate model and internal variability, both inherited from the larger spatial scales of the climate models (Table 1). The bias correction and spatial downscaling included with the downscaling methods aligns the climate model output to finer gridded observations but adds a relatively small portion to the overall uncertainty in MSD impacts. As shown in Figure 7, uncertainty due to downscaling is relatively small over Central America because the dominant sources of uncertainty come from large-scale climate model differences and internal variability. Downscaling methods primarily refine model output to match finer observational data but do not significantly increase the total uncertainty. Thus, in the context of MSD analysis, the role of downscaling in uncertainty is modest compared to inherited uncertainties from the climate models themselves. The MSD definition has the smallest contribution to total uncertainty at all warming levels with smaller contributions toward the Pacific coast where MSD frequency is greatest (Figure 8) and intensity is strongest (Figure 3)."

New references cited:

Alfaro, E. J.: Caracterización del "veranillo" en dos cuencas de la vertiente del Pacífico de Costa Rica, América Central, Rev. Biol. Trop. J. Trop. Biol. Conserv., 62, 1–15, https://doi.org/10.15517/rbt.v62i4.20010, 2014.

Karnauskas, K. B., Seager, R., Giannini, A., and Busalacchi, A. J.: A simple mechanism for the climatological midsummer drought along the Pacific coast of Central America, Atmósfera, 26, 261–281, 2013.

Perdigón-Morales, J., Romero-Centeno, R., Pérez, P. O., and Barrett, B. S.: The midsummer drought in Mexico: perspectives on duration and intensity from the CHIRPS precipitation database, Int. J. Climatol., 38, 2174–2186, 2018.

Zhao, Z., Han, M., Yang, K. and Holbrook, N.J.: Signatures of midsummer droughts over Central America and Mexico. Clim Dyn 60, 3523–3542, <a href="https://doi.org/10.1007/s00382-022-06505-9">https://doi.org/10.1007/s00382-022-06505-9</a>, 2023.

2. The current approach to perturbing the MSD definition seems somewhat arbitrary. I would like the authors to provide some additional context regarding these alternate definitions. For example, does shifting the window by 14 days make MSD impacts more/less relevant for different types of agriculture in the region? Or do these reflect different definitions used the existing literature? More broadly, why is shifting the window preferred over making changes to the minimum duration or minimum intensity, which seem at least as important?

This indeed was not well described in the original manuscript. We have redone the analysis adding to MSD timing multiple minimum intensity values to expand the variability of the MSD definition. The revised text provides a justification for the selection, which build on the findings of Maurer et al. (2022). See the response to comment 1 for the changes that are proposed for the manuscript.

3. In the paragraph beginning on line 71, I do not think that the impact definition should be conflated with uncertainty in its simulation across (e.g.) different hydrologic models. For example, when simulating hydrologic or agricultural drought, different hydrologic models will show varying responses for the same drought definition, while different drought definitions will manifest as an additional uncertainty source for each individual model. I think it would be worth more explicitly separating these sources of uncertainty in this paragraph.

To address this concern, we propose revising the paragraph to read:

When expanding an analysis to include specific impacts, varying definitions of impacts will add to the total uncertainty. For example, for future projections of potential evaporation (PE) for France, Lemaitre-Basset et al. (2022) found the PE formulation had a minor contribution to total projection uncertainty, except when only a single scenario was used. How droughts were characterized for compound hot and dry events was a dominant uncertainty source for low precipitation events but was a much smaller portion of uncertainty for other formulations (Jha et al., 2023). Even when given identical input, different models will simulate different impacts, compounding the uncertainty in projections (Chegwidden et al., 2019; Clark et al., 2016). The importance of this level of uncertainty can vary widely, based on the specific impact assessed (Bosshard et al., 2013).

**Minor points:**

1. Line 88: Looks like a missing word: "A recent study MSD explored the variability..."

**That has been corrected to "A recent study of the Central American MSD..."**

2. Line 114: I would suggest changing "climate model precipitation trends of the climate models" to "precipitation trends of the climate models."

**That has been revised as suggested.**

3. Line 110: It may be worth specifying that you are using the CMIP6 version of the NASA-NEX dataset, given there is also a CMIP5 version.

**Revised as suggested.**

4. Line 120,121: The SSP scenario should be 5-8.5, not 5-85.

**Corrected.**

5. In Figure 3, do the historical values come from the historical climate model simulations, rather than observational data? It would be worth stating this explicitly.

The caption for Figure 3 has been revised to include "Historical values as simulated by the model ensemble of MSD..."

6. Figure 6 is instructive but I think would benefit from showing the marginal distributions as KDEs along each axis.

We have attempted to modify the plot as suggested but have not been successful in adding this feature. We hope the manuscript can stand without this revision.

**Reviewer 2**

This manuscript analyzes uncertainty partitioning for the Central American midsummer drought (MSD) using multimodel ensembles, two downscaling datasets, and multiple MSD definitions evaluated at specific global warming levels. Its novel contribution is the explicit inclusion of MSD definition as an additional source of uncertainty, and it also demonstrates the value of applying a warming-level framing. This approach has been established in recent climate assessments but is applied here to hydrologic uncertainty partitioning to reduce sensitivity to model selection or emissions scenarios.

The study is a valuable contribution within the scope of HESS, addressing a climatically and socio-economically vulnerable region while applying an uncertainty framework relevant to decision-oriented science. The analysis is scientifically sound and the conclusions are well supported. With improvements to presentation clarity (streamlined writing, clearer figure formatting, consolidated summaries of uncertainty sources) and stronger connection to prior literature, the paper would make a strong contribution to HESS. After addressing the comments below, I would consider the scores across all criteria to be excellent.

**Thank you for your encouraging comments.**

**1. Scientific Significance**

The manuscript addresses a relevant and pressing hydrological issue in a vulnerable region and employs an uncertainty-partitioning framework aligned with HESS's interest in decision-relevant science. Its novel contribution is the explicit inclusion of the definition of the midsummer drought (MSD) as an additional quantified source of uncertainty, assessed alongside climate model spread, internal variability, and downscaling. The conclusion strengthens the broader relevance by noting that this framework can serve as a template for other precipitation-driven cases such as monsoons or rainy season timing (p.14, L285–287).

**Thank you for your encouraging comments.**

**Suggested enhancement**: The introduction notes that "*The uncertainty associated with each step...can become a daunting task for stakeholders*" (p.2, L30–33), but this theme is not revisited later. The discussion would be stronger if the authors explicitly linked their findings to future work aiming to be more decision-ready. In particular, they could emphasize two key contributions of their analysis: (i) that while model spread and internal variability dominate, the role of MSD definition is comparatively small, which highlights the need for future studies to continue testing how event or season definitions influence uncertainty; and (ii) that using specific warming levels rather than fixed time windows revealed how uncertainty evolves with global warming in a way that is less sensitive to model selection or emissions scenarios. Explaining why these findings are relevant for orienting future work toward decision-relevant applications would sharpen the paper's significance.

We thank the reviewer for their thoughtful comments and agree that the contributions of our findings could be highlighted further. We had noted on lines 269-270 that 'Despite having a strong impact on the magnitude and spatial extent of the MSD, the exact definition of the MSD has a minor effect on the uncertainty of MSD projections at all warming levels.' At the suggestion of the reviewer we now added: 'Thus, while model spread and internal variability dominate, the role of the MSD definition is comparatively small, which highlights the need for future studies to continue testing how event or season definitions influence uncertainty.'

Regarding the second point, we had noted (original lines 274 - 281) that 'The main implication of these findings for future work on climate disruption and the future of the Central American MSD is that selecting an ensemble of climate models is essential for characterizing the uncertainty in precipitation and its impact on the MSD. By analyzing impacts at specific levels of warming, rather than future spans of years, the selection of models may be done without excluding models based on sensitivity, which simplifies the process as other climate model skill metrics may be used. Using a single precipitation downscaling method for all climate models would still capture the majority of MSD impact uncertainty, though with multiple archives of

downscaled data freely available, multiple methods can be readily included. The definition of the MSD can be chosen to capture impacts of interest, but the specific definition of the time windows for the MSD does not add substantially to the uncertainty in impacts.' At the suggestion of the reviewer we now added that 'These findings show that future work can better support more efficient decision-making by selecting climate model ensembles based on performance metrics rather than sensitivity, framing projections by warming levels instead of time periods, and using a single downscaling method without major loss of uncertainty characterization. Additionally, they demonstrate that flexibility in defining the MSD allows tailoring analyses to stakeholder-relevant impacts without significantly increasing uncertainty.'

**2. Scientific Quality**

The study applies a sound methodological framework and delivers well-supported conclusions. However, certain aspects could benefit from deeper explanation or clarification:

**• Comparison with prior studies:**

The introduction cites studies where impact definition uncertainty was substantial (e.g., Lemaitre-Basset et al., 2022; Jha et al., 2023; Jeantet et al., 2023). Since this study finds MSD definition is a minor contributor, the discussion or conclusions should briefly contrast this outcome with those earlier findings.

We thank the reviewer for their comment. We state in our introduction that the importance of the level of uncertainty can vary widely, based on the specific impact assessed. We then name three examples, all three of which actually suggest that the specification had a varying or minor impact on the uncertainty (i.e. 'Lemaitre-Basset et al. (2022) found the PE formulation had a minor contribution to total projection uncertainty', 'How droughts were characterized for compound hot and dry events was a dominant uncertainty source for low precipitation events but was a much smaller portion of uncertainty for other formulations', 'Hydrology model parameterization did not significantly influence total uncertainty'), though the original paragraphs have been modified in response to comment 3by Reviewer 1. Thus our findings are actually in line with these prior studies. To clarify, we extended our existing sentence to now read: 'Despite having a strong impact on the magnitude and spatial extent of the MSD, the exact definition of the MSD has a minor effect on the uncertainty of MSD projections at all warming levels, similar to the variable or limited impact definition uncertainty of prior studies (Jeantet et al., 2023; Jha et al., 2023; Lemaitre-Basset et al., 2022).

**• References:**

Some statements require clearer sourcing. For example (L96–98), "In Nicaragua, distinctly precarious socio-economic and climatic vulnerabilities intersect with a scarcity of observational (station) data, rendering advances in the understanding of the regional

hydrologic system particularly pertinent". Stewart et al (2021) is cited in the prior sentence, but this is not a primary source for these specific claims. The authors should provide a more appropriate primary citation (e.g., to studies documenting data scarcity or socio-economic vulnerability in Nicaragua).

While the study by Stewart et al. study we cite underscores that further understanding of the regional hydrologic system is particularly pertinent, we added another citation as a primary source for the intersection of a scarcity of observational data and socioeconomic and climatic vulnerabilities.. The sentence now reads: 'In Nicaragua, distinctly precarious socio-economic and climatic vulnerabilities intersect with a scarcity of observational (station) data (Girardin, 2024), rendering advances in the understanding of the regional hydrologic system particularly pertinent (Stewart et al., 2021).

**• Methods:**

Restricting the analysis to model runs common to both CIL and NASA-NEX is a sound way to separate downscaling differences from model selection effects. However, the characterization of internal variability may be limited by relying on single realizations per model, and equal model weighting may understate the effect of model dependence or skill. These limitations should be acknowledged.

We thank the reviewer for their comment. We now added the following sentence: 'It should be noted that restricting the analysis to model runs common to both CIL and NASA-NEX, with a single run per model, may limit the characterization of internal variability by relying on single realizations per model, and equal model weighting may understate the effect of model dependence or skill.'

**• Discussion depth:**

o The results show downscaling adds ~10–15% uncertainty, but the discussion does not explain why this is relatively small compared to other studies. A short explanation (e.g., method characteristics, regional features) would help.

We added the following explanation: 'As shown in Figure 7, uncertainty due to downscaling is relatively small over Central America because the dominant sources of uncertainty come from large-scale climate model differences and internal variability. Downscaling methods primarily refine model output to match finer observational data but don't significantly increase the total uncertainty. Thus, in the context of MSD analysis, the fraction attributed to downscaling in uncertainty is modest compared to inherited uncertainties from the climate models themselves.'

o Figures 7–8 could be discussed in more depth, particularly why duration and intensity differ in uncertainty contributions and why MSD definition plays a larger role for duration.

We added the following explanation: 'The uncertainty due to MSD duration is higher

than that for intensity because the thresholds for the start and end points depend strongly on the definition used to identify the MSD. Slight changes in the definition (e.g., a new minimum dry period) can shift the start and end dates, thus changing the duration. By contrast, while the definition still matters for intensity the actual reduction in rainfall during an MSD above a given minimum threshold is stable across different definitions.'

 Explicitly linking the dominance of model/internal variability back to implications for stakeholders would strengthen conclusions.

We added the following explanation in the conclusions: Furthermore, the dominance of internal variability at near-term timescales, and the growing role of model uncertainty under higher warming, has important implications for stakeholders. Policymakers, water managers, and agricultural planners must recognize that near-term variability may mask or amplify underlying trends, complicating adaptation strategies. In the longer term, the prominence of model uncertainty highlights the need for improved climate modeling and ensemble strategies to better constrain future risk assessments. Explicitly accounting for which source of uncertainty dominates at a given time horizon allows stakeholders to tailor their decisions accordingly.

 The introduction's explanation of why precipitation is harder to project than temperature is not revisited. Connecting this physical reasoning to the results would situate findings more broadly.

We amended the introduction to now read: 'They also observed a marked difference between precipitation projections, with greater internal and model variability persisting late into the 21st century, and temperature projections, which showed scenario uncertainty dominating projections in most regions late in the 21st century. This reflects the dominant physics of temperature being a primary response to the increased radiative forcing of accumulating greenhouse gases, and precipitation being driven by secondary physical processes that are more challenging to model, such as the moisture holding capacity of the atmosphere, the variety of phenomena that can cause precipitation, and feedbacks with the land surface, ocean, and cryosphere lead to significant variability on scales much smaller than those of temperature (Neelin et al., 2022; O'Gorman and Schneider, 2009; Stainforth et al., 2005). 'As the MSD is a tropical, precipitation-driven phenomenon, the difference in the difficulty of projection between temperature and precipitation has little importance for the results regarding uncertainty partitioning. We therefore did not make any changes to the conclusions.

**3. Presentation Quality**

The manuscript is structured and purposeful overall, but several presentation

enhancements would significantly improve readability and clarity:

Thank you for your suggestions.

**• Clarity of writing:**

Some sentences are overly long or disrupted by parenthetical clauses, making them hard to follow.

o For example, the sentence at lines 29–30 could be simplified to: "This study focuses on future precipitation-driven hydrologic changes, which introduce a cascade of uncertainties into impact projections (Aitken et al., 2023)."

**Thank you for this suggestion. We adopted this version.**

o Similarly, the sentence at lines 30–33 could be rephrased as: "Uncertainty arises at each step of this cascade—including future greenhouse gas concentrations, climate response, downscaling, and hydrologic response—and can be estimated using multimodel ensembles (discussed in more detail below). However, this estimation can be daunting for stakeholders preparing strategies to cope with projected changes in the timing and availability of water."

We simplified this sentence as given below (l. 30-33): The uncertainty associated with each step along this cascade, which can include future greenhouse gas concentrations, climate response, downscaling, and hydrologic response, can be estimated using multi-model ensembles (discussed in more detail below). Assessing this uncertainty can become a daunting task for stakeholders preparing strategies to cope with the projected changes in the timing and availability of water.

o Similar improvements can be made to other heavily clause-laden sentences.

We simplified another clause-laden sentence to read: 'Improved understanding of the comparative magnitudes of different sources of variability in impact projections can highlight opportunities to reduce them. Even more importantly, these comparative magnitudes can help identify which steps in the modeling chain may be simplified without adversely affecting metrics relevant to decision-making related to adaptation and mitigation strategies in water resources (Steinschneider et al., 2023).

This paragraph was also revised in response to this comment and also in response to Comment 1 by Reviewer 1. Please refer to that response to see the detailed revisions.

**Summary of uncertainties:**

The uncertainty sources are described in text but not consolidated. A clear listing or table early in the manuscript would help readers track sources and enhance structural clarity.

We thank the reviewer for their thoughtful observation. We now provide a listing of the uncertainty sources at the beginning of the Methods section that reads: 'The main sources of uncertainty in projections of the mid-summer drought (MSD) evaluated in this study include internal variability, differences among climate models, the choice of downscaling method,

and the definition of the MSD itself. They are determined based on climate projections of daily precipitation and the simulated MSD characteristics (the impacts of concern for this study) and described as follows.

**Linking introduction to conclusions:**

The conclusion would benefit from explicitly revisiting key themes from the introduction—stakeholders, simplification opportunities, warming-level framing—to improve narrative cohesion.

We agree with the reviewer and added the following paragraph to the end of the Conclusions section: 'This study directly responds to the needs of stakeholders, such as water managers and agricultural planners, who require actionable and skillful projections to inform adaptation strategies under climate change. By identifying where simplifications in modeling (e.g., MSD definition or downscaling method) do not substantially impact uncertainty, and by adopting a warming-level framing that aligns with international policy targets, this work supports more efficient and targeted planning in the face of future hydrologic change that can be developed for other geographic regions."

**• Definition of MSD metrics:**

MSD intensity should be defined unambiguously, e.g., "MSD intensity was defined as the mean precipitation of the two seasonal maxima minus the minimum precipitation between them (derived from the smoothed daily series)." Figure 2 should clarify that red points mark maxima and the blue point marks the minimum used, ideally through a caption note and a simple legend.

We thank the reviewer for their suggestion. This portion of the Methods section was revised extensively in response to comments by Reviewer 1. The revised paragraph now includes:

'For this experiment, we use the same definitions of intensity and duration as Maurer et al. (2022): MSD intensity was calculated as the mean precipitation of these maxima minus the minimum precipitation occurring between them; MSD duration was defined as the number of days between the two seasonal precipitation maxima.'

We agree with the reviewer about the clarifications for Fig. 2. The figure caption now reads: 'A schematic of a typical MSD year, highlighting the definition (dates) and the impacts (intensity and duration) of interest in this study. Red points mark maxima and the blue point marks the minimum used. Duration is the number of days between the peaks; intensity is the average of the two peaks minus the minimum between them. All metrics are calculated from the smoothed daily precipitation time series. To estimate the effect of the definition on MSD variability, dates are shifted 14 days earlier and later from the definition dates shown here.' We believe that the labels on the graphic and other revisions as noted below, together with the figure caption revised according to the reviewer's suggestion, clearly illustrate the different characteristics of the MSD, and do not

**necessitate an additional legend.**

**Notation for warming levels:**

Warming-level notation throughout the text and figures (Figures 1 and 8) should follow IPCC conventions, using a "+" sign before values (e.g., "+1.5 °C," "+2 °C," "+3 °C").

These changes have been made throughout the manuscript.

**Abstract:**

The abstract is informative, identifying which uncertainty sources dominate and provides guidance for future studies. To strengthen its impact, the authors could add a clearer statement of relevance for future work wishing to inform water planning/adaptation.

Thank you for this suggestion. In response, we added the following statement to the abstract: 'These findings provide critical guidance for future research aiming to inform water planning and adaptation efforts in the region: by identifying the dominant sources of uncertainty across warming levels, this framework helps prioritize where to focus modeling and monitoring efforts. In particular, water resource managers can use this information to design adaptive strategies that are robust to model spread and shifts in seasonal precipitation timing, rather than to definitional ambiguity. The projection uncertainty partitioning approach could serve as a template to quantify the relative importance of uncertainty for projections of other precipitation-driven phenomena in different geographic contexts.'

**Figures:**

o Ensure consistent font type and size (Arial or Helvetica recommended by HESS) across all figures.

o Figure 1: Add "+" to Y-axis ticks; ensure "2080" is fully visible.

**Revised as requested**

o **Figure 2:** Increase axis font size, add more x-axis ticks/labels, enlarge plotted blue and red points, clarify caption the caption to reference the blue and red data points, include a legend for the blue and red data points, fix cutoff of "0."

Revised as requested, except for two details: adding more x-axis labels made it too crowded, and we feel that mentioning the red and blue dots in the caption will be sufficient without adding a legend for the dots.

o **Figure 3:** Increase font size, and include a legend for CADC boundary, coastlines, and significance marks.

Font sizes were increased, and a legend was added as requested.

o **Figure 6:** Spell out "Probability," remove underscore in "Ensemble\_mean," improve font consistency.

**Revised as suggested.**

o **Figure 7:** Make color bar clearer with black border/ticks to be consistent with color bars in other figures; replace the legend title of "Percent" with "% of total variance" or "Contribution to total variance (%)."

Revised as suggested. The new Figure 8 of MSD frequency is also consistent with this revised figure.

o **Figure 8:** Remove underscore from "MSD\_definition," use a colorblind-friendly palette (e.g., Blue #0072B2, Orange #E69F00, Sky Blue #56B4E9, Vermilion #D55E00).

Note that this is now Figure 9. It has been revised as suggested.